# Screening of Small-Molecule Libraries Using SARS-CoV-2-Derived Sequences Identifies Novel Furin Inhibitors

**DOI:** 10.3390/ijms25105079

**Published:** 2024-05-07

**Authors:** Alireza Jorkesh, Sylvia Rothenberger, Laura Baldassar, Birute Grybaite, Povilas Kavaliauskas, Vytautas Mickevicius, Monica Dettin, Filippo Vascon, Laura Cendron, Antonella Pasquato

**Affiliations:** 1Department of Pharmaceutical and Pharmacological Science, University of Padova, Via Marzolo, 5, 35131 Padova, Italy; alireza.jorkesh@studenti.unipd.it; 2Department of Biology, University of Padua, Viale G. Colombo 3, 35131 Padova, Italy; filippo.vascon@unipd.it (F.V.); laura.cendron@unipd.it (L.C.); 3Institute of Microbiology, University Hospital Center and University of Lausanne, Rue du Bugnon 48, 1011 Lausanne, Switzerland; sylvia.rothenberger-aubert@chuv.ch; 4Spiez Laboratory, Federal Office for Civil Protection, Austrasse, 3700 Spiez, Switzerland; 5Department of Industrial Engineering, University of Padova, Via Marzolo 9, 35131 Padova, Italy; laura.baldassar@studenti.unipd.it (L.B.); monica.dettin@unipd.it (M.D.); 6Department of Organic Chemistry, Kaunas University of Technology, Radvilenu Rd. 19, LT-50254 Kaunas, Lithuania; birute.grybaite@ktu.lt (B.G.); vytautas.mickevicius@ktu.lt (V.M.); 7Joan and Sanford I. Weill Department of Medicine, Weill Cornell University, 1300 York Avenue, New York, NY 10065, USA; 8Biological Research Center, Lithuanian University of Health Sciences, Tilzes Str. 18, LT-47181 Kaunas, Lithuania; 9Institute of Infectious Diseases and Pathogenic Microbiology, Birstono Str. 38A, LT-59116 Prienai, Lithuania; 10Department of Microbiology and Immunology, University of Maryland School of Medicine, 655 W. Baltimore Street, Baltimore, MD 21201, USA

**Keywords:** Furin, protease, SARS-CoV-2, cleavage, inhibitor, *in vitro*, HTS, envelope glycoprotein, virus, exosite, peptide

## Abstract

SARS-CoV-2 is the pathogen responsible for the most recent global pandemic, which has claimed hundreds of thousands of victims worldwide. Despite remarkable efforts to develop an effective vaccine, concerns have been raised about the actual protection against novel variants. Thus, researchers are eager to identify alternative strategies to fight against this pathogen. Like other opportunistic entities, a key step in the SARS-CoV-2 lifecycle is the maturation of the envelope glycoprotein at the RARR685↓ motif by the cellular enzyme Furin. Inhibition of this cleavage greatly affects viral propagation, thus representing an ideal drug target to contain infection. Importantly, no Furin-escape variants have ever been detected, suggesting that the pathogen cannot replace this protease by any means. Here, we designed a novel fluorogenic SARS-CoV-2-derived substrate to screen commercially available and custom-made libraries of small molecules for the identification of new Furin inhibitors. We found that a peptide substrate mimicking the cleavage site of the envelope glycoprotein of the Omicron variant (QTQTKSHRRAR-AMC) is a superior tool for screening Furin activity when compared to the commercially available Pyr-RTKR-AMC substrate. Using this setting, we identified promising novel compounds able to modulate Furin activity *in vitro* and suitable for interfering with SARS-CoV-2 maturation. In particular, we showed that 3-((5-((5-bromothiophen-2-yl)methylene)-4-oxo-4,5 dihydrothiazol-2-yl)(3-chloro-4-methylphenyl)amino)propanoic acid (**P3**, IC_50_ = 35 μM) may represent an attractive chemical scaffold for the development of more effective antiviral drugs via a mechanism of action that possibly implies the targeting of Furin secondary sites (exosites) rather than its canonical catalytic pocket. Overall, a SARS-CoV-2-derived peptide was investigated as a new substrate for *in vitro* high-throughput screening (HTS) of Furin inhibitors and allowed the identification of compound **P3** as a promising hit with an innovative chemical scaffold. Given the key role of Furin in infection and the lack of any Food and Drug Administration (FDA)-approved Furin inhibitor, **P3** represents an interesting antiviral candidate.

## 1. Introduction

The year 2020 will be remembered in history as a pandemic year. Contrary to the prevailing consensus among leading virologists, who had anticipated the emergence of a deadly influenza virus [1], in the last four years, severe acute respiratory syndrome coronavirus 2 (SARS-CoV-2) has spread all over the world. The initial wave of infection hit the entire global population like a tsunami, claiming hundreds of thousands of victims in a very short period [2,3]. The disease caused by SARS-CoV-2 is coronavirus disease 2019 (COVID-19). The symptoms can vary in severity from none at all (asymptomatic) to fever, cough, sore throat, general weakness, fatigue and muscle pain, and loss of smell and taste. The most severe cases can lead to shortness of breath due to pneumonia and acute respiratory distress syndrome as well as other complications, potentially leading to death (https://www.ecdc.europa.eu/en/covid-19, access on 1 March 2024) [4]. The turning point arrived with the advent of vaccines, developed at an unprecedented pace and found to possess high efficacy in preventing severe COVID-19 illness and death [5]. Despite the success of vaccination, many researchers are still engaged in the identification of suitable drugs against SARS-CoV-2. Indeed, the virus does not stop mutating, and ongoing viral evolution gives rise to new strains, with possible future potential to escape from vaccine-induced immunity (https://www.ecdc.europa.eu/en/covid-19/variants-concern, access on 1 March 2024) [6,7].

A crucial step in the viral lifecycle is the maturation of the envelope spike (S) glycoprotein into the S1 and S2 subunits. Differently from other coronaviruses of the same clade, the SARS-CoV-2 spike relies on the cellular enzyme Furin for cleavage at the peculiar multi-basic motif NSP**RR**A**R**_685_↓ [8]. Furin is a member of the proprotein convertase family and acts as a regulator of several proteins that gain activity or, vice versa, are degraded by Furin processing, typically at clusters of basic amino acids [9,10]. The exploitation of this cellular enzyme for the cleavage of the envelope glycoprotein is quite a common event in the viral world, e.g., Ebola virus, highly pathogenic influenza viruses, and Human Immunodeficiency Virus (HIV-1) are well-known Furin-dependent pathogens [10,11]. With time, most SARS-CoV-2 variants that spread had acquired additional positive charges around the scissile bond, e.g., the Omicron N679K, P681H variant, bearing the **K**S**HRR**A**R**_685_↓ motif. These Omicron mutations confer a gain-of-function phenotype to the virus since the kinetics of maturation of the spike glycoprotein is faster [12,13]. Spike cleavage plays a crucial role in SARS-CoV-2 infectivity by releasing the “fusion peptide” located at the N-terminus of S2 [14]. This step is essential as it unlocks the fusion process, which is vital for the virus to enter target cells [15]. Thus, no Furin-independent variant has ever been reported (https://gisaid.org/hcov19-variants/, access on 1 March 2024) [16,17,18]. Recent findings further suggest that the SARS-CoV-2 Spike is capable of inducing cell-cell fusion, bypassing its receptor ACE2. This activity can be impaired by Furin inhibitors [19]. As a consequence, the processing site has become an ideal drug target to block infection, so that several compounds have been proposed to interfere with Furin activity and therefore viral infection [20,21]. Scientific efforts in this direction are important in order to provide valid alternatives to other therapeutic strategies. Indeed, the wide spread of SARS-CoV-2 has prompted the emergence of multiple escape variants that are able to evade not only small molecules but also the antibody defense gained by vaccine immunization [22].

A huge boost in research on Furin inhibitors has occurred in the last few years due to the pandemic. Nonetheless, there have been no Food and Drug Administration (FDA)-approved Furin inhibitors to date. One line of research has focused on the development of antibodies targeting either the viral spike cleavage site [23,24] or Furin [25]. However, a major limitation of this approach is the correct delivery of such bulky molecules that cross the cell membrane in order to reach the Golgi stack where Furin works. Historically, Furin inhibitors have been clustered into three different categories: protein based, peptide-based, and small molecules. For the sake of brevity, here, we give only a few examples. α1-Antitrypsin Portland (α1-PDX) is a protein-based Furin inhibitor that was bioengineered from the α1-antitrypsin serpin. α1-PDX blocks the processing of HIV-1 and measles virus envelope glycoproteins and therefore viral spread [26]. Accordingly, α1-PDX antagonizes COVID-19 as well [27]. Peptide-based inhibitors represent an attractive alternative to large molecules because they are cheaper and easier to synthesize and deliver. Decanoyl-RVKR-chloromethylketone (dec-RVKR-cmk, CMK) is the gold standard *in vitro* Furin inhibitor. CMK blocks the activation of different viral glycoproteins [28] including SARS-CoV-2 envelope S glycoprotein [29]. More recently, various peptidomimetic inhibitors have been developed against Furin to overcome the intrinsic high toxicity of CMK. Among these, MI-1851 has been found to potently prevent SARS-CoV-2 S protein cleavage, significantly decreasing viral titers from infected cells [30]. Finally, a plethora of different small molecules have been proposed as effective SARS-CoV-2 antivirals by targeting Furin. A very promising compound is BOS-318, which, differently from the majority of the other Furin inhibitors, does not target the catalytic site but a side groove on the molecule surface. BOS-318 is highly selective against Furin (IC_50_ = 1.9 nM); it is cell-permeable and was developed as an effective treatment for cystic fibrosis airway disease [31]. Tested in the context of SARS-CoV-2 infection, BOS-318 has been found to be effective in containing viral cell-to-cell spread as well [20].

Looking for novel inhibitors of Furin activity is important since these molecules may provide effective drug treatments against major human pathologies [9], including COVID-19 [32]. A major drawback of the currently available Furin inhibitors is the lack of substrate specificity; that is, the inhibitors block all Furin activities regardless of the nature of the substrate. This is an issue because the enzyme is involved in a plethora of physiological cell functions besides the processing of viral glycoproteins. Thus, keeping Furin host activities intact while blocking pathogen maturation is of high priority. In this context, the choice of the substrate is very important since we [12,33,34] and others [35] have found that the identity of the amino acids surrounding the scissile bond is crucial for inferring higher/lower cleavability. Accordingly, the potency of an inhibitor may vary when tested against different substrates of the same enzyme. Considering this context, our team has developed an assay to facilitate the *in vitro* screening of novel Furin inhibitors that target the cleavage site of the SARS-CoV-2 envelope glycoprotein, utilizing a uniquely designed fluorogenic peptide, SYQTQTKSHRRAR-(7-Amido-4-methylcoumarin) [AMC]. This peptide, an innovative construct proposed by our research group and mimicking the Omicron variant cleavage site of SARS-CoV-2 envelope glycoprotein, represents the first use of this sequence in such applications. Indeed, typical *in vitro* Furin activity assays employ the Pyroglutamic(Pyr)-RTKR-AMC peptide [36], which is much shorter and bears a different amino acid sequence. As a matter of fact, Pyr-RTKR-AMC is the gold standard reference used for Furin inhibitor screening, e.g., [37,38]. Here, we successfully replaced Pyr-RTKR-AMC with the new SARS-CoV-2-derived Furin substrate SYQTQTKSHRRAR-AMC, showing that the latter is suitable for Furin high-throughput screening (HTS). Furthermore, proof-of-concept assays using commercially available and custom-made compound libraries have identified a novel Furin inhibitor that can block the processing of the viral-derived substrate more efficiently than that of Pyr-RTKR-AMC.

In summary, Furin inhibitor research has surged due to the pandemic, yet no FDA-approved inhibitors exist. Specificity remains a critical goal due to the numerous cellular functions of the enzyme that should be preserved while targeting of viral glycoprotein cleavage should be preferred. Our research has introduced a new SARS-CoV-2-derived Furin substrate for high-throughput screening, offering a promising avenue for the discovery of more efficient Furin inhibitors.

## 2. Results

### 2.1. Omicron SARS-CoV-2-Derived Peptide Is a Superior Furin Substrate

As of spring 2023, the most diffused SARS-CoV-2 variant was Omicron, which carries multiple mutations within the envelope S glycoprotein. Notably, two of them—N679K and P681H—emerged right at the Furin cleavage site. We [12] and others [13] have shown that these mutations confer a gain-of-function (GOF) phenotype. Thus, the peptide SYQTQTKSHRRAR_685_↓SVAS is digested much faster than the WT version and at different pH values.

The gold standard substrate for the *in vitro* Furin activity test is Pyroglutamic-RTKR-AMC (standard peptide). Based on the observation that SARS-CoV-2-derived substrates are very sensitive to Furin, we engineered the peptide SYQTQTKSHRRAR-AMC (SARS peptide) (Figure 1) to investigate whether the viral sequence could represent a better alternative to the standard peptide for testing Furin activity *in vitro* and in high-throughput screening (HTS).

C-terminally AMC-labeled peptides are very popular as protease reporter substrates since processing can be easily monitored by fluorescence intensity measurements [33]. When bound to the peptide, AMC fluorescence is quenched; thus, AMC release by protease-mediated cleavage increases its fluorescence at λ_ex_ = 360 nm/λ_em_ = 460 nm. The released fluorescence is proportional to the propensity of the peptide to processing. First, we verified the cleavability of SARS peptide by incubating increasing concentrations (0.01, 0.10, 1, 5, 10, 20, 50, and 100 μM) of substrate with soluble human Furin (sFur) at different pH values (5.5, 6.5, 7.0, and 7.5). As expected, the substrate was hydrolyzed by the enzyme, and higher peptide concentrations matched higher fluorescence and initial reaction rates, confirming a Michaelis–Menten behavior. Neutral/slightly basic pH values favored the cleavage of the SARS peptide (higher V_MAX_ and lower K_m_; Figure 2A, Table 1). Next, we validated the assay by using the well-known Furin-specific Decanoyl-RVKR-chloromethylketone (CMK) inhibitor. Briefly, sFur was incubated either with CMK or dimethylsulfoxide (DMSO, control) for 5 min prior to adding 2.5 μM SARS peptide (Figure 2B).

After 1 h, the fluorescence was read, and the Z score was calculated to be 0.65, indeed suggesting that the SARS peptide can be used in an HTS setting [39]. Finally, we tested side-by-side the cleavability of SARS vs. standard peptides to determine which substrate performs better *in vitro*. At all pH conditions tested, SARS peptide was found to be a superior substrate (Figure 3). Interestingly, we noted that the fluorescence released immediately after the addition of the enzyme was significantly higher in the case of SARS peptide when the pH was >6.5. These data are in line with our previously reported results where the entire envelope glycoprotein showed a peculiar kinetic of cleavage with a very high rate of conversion into S1/S2 at very early time points [12].

Overall, a 13mer peptide mimicking the cleavage site of the envelope S glycoprotein of SARS-CoV-2 and carrying a C-terminal AMC group represents an excellent substrate to test *in vitro* the activity of the cellular Furin protease.

### 2.2. Screening of Small-Compound Libraries Using the SARS Peptide

Our results suggest that the SARS peptide is an effective tool for HTS. Thus, as proof-of-concept, the substrate was used to search for novel inhibitors of Furin in small-scale reactions set up in 96-well plates. Any drug that can interfere with this cellular enzyme is of interest, given the role played by Furin in SARS-CoV-2 infection as well as other pathologies. The BCDP drug-like small-molecule library (BCDP library) was kindly provided by P. Kavaliauskas from Weill Cornell Medicine of Cornell University (Povilas library, Appendix A). The HTS entailed sFur pre-incubation with either the drug (Prestwick compounds at 10 μM; Povilas compounds at concentrations detailed in the Materials and Methods) or DMSO (negative control). CMK, a potent Furin inhibitor, was included as a positive control (1 μM). After 5 min, 5 μM SARS substrate was added, and the fluorescence was recorded over 1 h (λ_ex_ = 360 nm/λ_em_ = 460 nm). Activity was calculated as the relative fluorescence unit (RFU) increase compared to the negative control (considered as 100% activity; Figure 4). Compounds capable of lowering Furin activity > 50% were considered as a hit (Appendix A). No potent inhibitors stemmed from the commercially available library (Figure 4).

No potent inhibitors pop out from the commercially available library (Figure 4). In contrast, the custom-made library screening resulted in several potential novel inhibitors (Appendix A). Among these, only the drugs that did not interfere with fluorescence measurement at λ_ex_ = 360 nm/λ_em_ = 460 nm (**P2**–**P18**, Appendix A) were taken into consideration and went through the next validation step, consisting of testing the compound inhibitory potencies in the 2.5–100 μg/mL range. Within this group, six drugs—namely, **P3**, **P5**, **P7**, **P9**, **P13**, and **P16**—were confirmed as novel Furin inhibitors, showing typical dose-dependent activity, while the rest of the potential hits were discarded (Figure 5).

The calculated IC_50_ values were 35 μM for **P3**, 62 μM for **P7**, and 79 μM for **P16**, respectively. The other three compounds, despite being able to block Furin activity, were less effective and did not follow the typical S-shaped (sigmoidal) pattern of inhibitory enzyme kinetics. Interestingly, **P3**, **P7,** and **P16** possess similar chemical scaffolds, whereas the other three, namely, **P5**, **P9**, and **P13**, have a chemically different identity. Specifically, **P3** and **P16** are identical with the exception of an extra bromine atom attached to the thiophene heterocycle at position 2 in **P3** (Figure 5 and Figure 6). **P7** is somehow different, being characterized by a similar scaffold but with the thiophene replaced by a dimethyl phenylamine group attached to the thiazolone moiety and the 3-chlorotoluene replaced by a chlorobenzene (Figure 5 and Figure 6B).

In summary, *in vitro* screening of small molecules using the SARS peptide substrate identified six different novel Furin inhibitors. Of note, these compounds possess original chemical scaffolds that have never been reported before among the known Furin inhibitors.

### 2.3. Characterization of the Furin Inhibitor **P3**

**P3** inhibits SARS peptide processing by soluble Furin with an IC_50_ of 35 μM (Figure 4). In order to understand the specificity of this small compound, we tested its ability to interfere with the processing of the gold standard Pyr-RTKR-AMC peptide. The latter is routinely used for *in vitro* screening libraries against Furin enzymatic activity [38]. In order to fairly compare the inhibition of the two substrates, we ran a new set of experiments using the same conditions as described in Figure 4 but with Pyr-RTKR-AMC as a substrate. Briefly, sFur was pre-incubated with an increasing amount of **P3** up to 100 μM. Following the addition of Pyr-RTKR-AMC, we monitored the release of the AMC fluorescent group over time. Interestingly, we found that the potency of **P3** was poorer when compared to the ability of the very same inhibitor to block SARS peptide cleavage (Figure 7A).

Next, we focused on the closely related subtilisin kexin isozyme-1 (SKI-1), also known as site 1 protease (S1P). Furin and SKI-1/S1P belong to the same family, the Proprotein Convertases (PCs), but they have distinct consensus cleavage sequences. Typically, Furin cleaves after dibasic residues, whereas SKI-1/S1P processes at RX(L/V/I)X↓ [9]. On the blueprint of the above reported tests, soluble SKI-1/S1P (sSKI-1/S1P) [34] was incubated with the **P3** molecule prior to adding the Ac-IYISRRLL-AMC substrate [33]. We found no significant inhibitory effect (Figure 7B), thus confirming there is no cross-reactivity.

Overall, the **P3** inhibitor potently blocks SARS-derived SYQTQTKSHRRAR-AMC processing by Furin, while showing very modest inhibitory activity on Furin-mediated cleavage of (Pyr-RTKR-AMC) and a rather absent effect on SKI-1/S1P.

### 2.4. Possible Mechanism of Action of the **P3** Furin Inhibitor

Our data suggest that **P3** is not as potent against the standard peptide as the SARS-derived peptide under the same conditions. In order to better characterize its mode of action of how the inhibitor works, we used bioinformatic tools to gain more information on the **P3**/Furin interaction. We took advantage of the available crystal structure of Furin (PDB id, 4Z2A) to identify likely druggable grooves on the surface of the enzyme in addition to the obvious catalytic pocket. Using FTPocketWeb 1.0.1 (https://durrantlab.pitt.edu/fpocketweb/, default parameters, access 15 November 2023) [40], we identified three major exosites (Pockets 1–3, score −7.025, −7.006, and −7.107, respectively) with high druggability. Pocket 1 (light green) and 2 (orange) sit close to each other, and they are located on the opposite side of the Furin surface when compared to the major catalytic pocket (dark green-violet; Figure 8). Nearby the latter, the server highlighted an additional distinct groove, namely, Pocket 3 (cyan; Figure 8).

Next, we investigated whether **P3** possesses any affinity towards either the Furin catalytic site or any of the identified exosites using a preferential docking approach (AutoDock 4.2.6 [41] (score −7.025 kcal/mol, −7.006 kcal/mol, and −7.107 kcal/mol, for Pockets 1, 2, and 3, respectively) and AutoDock Vina v1.2.5 score −6.07 kcal/mol, −6.65 kcal/mol, and −7.55 kcal/mol, for Pockets 1, 2, and 3, respectively) [42,43], see the Materials and Methods for details). The analysis revealed that the small molecule may interact with Pocket 3 (Figure 9A), suggesting that its mechanism of action may not rely on a direct competition with the substrate for the catalytic pocket. Rather, **P3** may somehow function through the interference between the amino acids surrounding the cleavage site of SYQTQTKSHRRAR-AMC and the Furin surface. Specifically, the **P3** functional groups that engaged in specific interactions with the protease are the aromatic thiophene and toluene in addition to the carboxylic group. The fact that these chemical motifs are present also in the other newly identified inhibitors suggests that a common scaffold mediates the inhibitory activity. From the protease point of view, the residues involved are Trp531, Val263, Arg490, Ala532, and Asp264 (Figure 9B).

Notably, using a blind-docking approach (Achilles Blind Docking Server https://bio-hpc.ucam.edu/achilles/, access on 14 October 2023), **P3** ended up fitting into the same Furin surface Pocket 3 (Figure 10A), further supporting the likely ability of this compound to target this specific enzyme spot. Importantly, the blind docking results were validated by similar analyses conducted on BOS-318 (Figure 10B), a well-described Furin inhibitor known for binding to a distinctive enzyme exosite [31].

These findings all indicate that the primary interaction of **P3** with Furin may occur in the exosite Pocket 3 rather than in the catalytic site. The interaction, which does not touch the enzymatic core of the protease, may be key to understanding the ability of **P3** to inhibit Furin in a substrate-specific manner. While molecular docking provides a static snapshot of the interaction between the enzyme and the inhibitor, molecular dynamics offer a more comprehensive view, revealing continuous interaction details over time. Accordingly, in forthcoming experiments, we plan to use molecular dynamics to gain a better understanding of the mechanism of action of this inhibitor.

## 3. Discussion and Conclusions

A common hallmark of some highly pathogenic viruses, including Ebola virus, is their strict dependency from the host protease Furin to attain full maturation of their surface glycoprotein. This is also the case of SARS-CoV-2, which—as soon as its envelope glycoprotein acquired a multibasic motif [18]—turned into a threat to the entire human population. Intriguingly, the processing by Furin seems to be essential, as no SARS-CoV-2-escape variants have ever been detected so far, in spite of massive random mutations that have occurred within the spike S protein. Of note, during early spread, the pathogen showed a marked propensity to refine those amino acids located all around the S1/S2 boundary, without altering the RARR_685_↓ motif. Specifically, the most popular Omicron variant bears the N679K, P681H replacements that we [12] and others [13] have shown to confer a gain-of-function. The presence of additional positive charges beyond the strictly conserved RRAR motif significantly enhances the cleavage of the envelope glycoprotein. Interestingly, although the amino acids at positions 679 and 681 do not directly contact the catalytic pocket of the enzyme, their specific identities can substantially influence the rate of processing. This effect is likely mediated through interactions with the surrounding surface area of Furin. The realization that protagonists of the cleavage extend beyond the 2–4 basic residues near the scissile bond and the catalytic pocket is crucial. Indeed, understanding that the surrounding amino acids can influence cleavage provides researchers with an additional strategy to interfere with the process, either by directly interacting with these distant amino acids or by subtly altering the local protease conformation to induce allosteric effects. Building on these observations, we designed an extended fluorogenic substrate—SYQTQTKSHRRAR-AMC (SARS peptide)—for *in vitro* Furin activity assays and inhibitor screenings. This sequence was found to possess excellent cleavability [12], thus representing an appealing replacement for the canonical Pyr-RTKR-AMC peptide. The superiority of viral-derived sequences in being processed by host proteases has a precedent. As an example, the Lassa virus-derived peptide IYISRRLL is by far the best substrate for the human proprotein convertase SKI-1/S1P [33]. Therefore, our studies further encourage the use of viral-derived sequences for the engineering and development of sensitive *in vitro* enzymatic assays. As a matter of fact, the use of a better cleavable substrate may allow actual minimization of the chemicals needed for the tests. Since HTS is normally performed over an extended collection of compounds, the screening may be more cost effective. In addition, another aspect deserves to be highlighted here: The differences between SARS-CoV-2 and standard Pyr-RTKR-AMC digestions stress the intrinsic dissimilar nature among substrates. Does a general sequence represent a good surrogate substrate for the identification of inhibitors against enzymes? Rather, would the use of a specific sequence be more effective for developing novel compounds to switch off specific processing but not others?

The use of the novel SARS-CoV-2-derived substrate, coupled with the innovative, non-commercially available library of small compounds provided by Dr. Povilas Kavaliuskas, represents a pioneering approach in Furin inhibition studies. This unique combination of resources has yielded remarkable results, surpassing expectations by yielding multiple hit compounds. This success stands in stark contrast to conventional methodologies reliant on standard substrates and commercially available libraries. By breaking away from traditional paradigms, we have unlocked new avenues for discovery, showcasing the potential for groundbreaking advancements in drug development and molecular research.

Proof-of-concept screenings have revealed that the sensitivity of Furin towards potential inhibitors depends on the exact aminoacidic sequence of the reporter substrate. For example, quercetin [37] is more effective in blocking the cleavage of the classical rather than the SARS-CoV-2-derived peptide. The most interesting data were retrieved from the survey of a custom-made library, which is characterized by a collection of unique and novel chemical scaffolds (Povilas Library). From a pool of roughly 300 molecules, six emerged as potential Furin inhibitors, specifically inhibiting SYQTQTKSHRRAR-AMC substrate processing. The most potent one—3-((5-((5-bromothiophen-2-yl)methylene)-4-oxo-4,5-dihydrothiazol-2-yl)(3-chloro-4-methylphenyl)amino)propanoic acid, referred to as **P3**—possesses an IC_50_ of 35 μM. Two others—3-((4-chlorophenyl)(5-(4-(dimethylamino)benzylidene)-4-oxo-4,5-dihydrothiazol-2-yl)amino)propanoic acid, named **P7**, and 3-((3-chloro-4-methylphenyl)(4-oxo-5-(thiophen-2-ylmethylene)-4,5-dihydrothiazol-2-yl)amino)propanoic acid, named **P16**—were able to block Furin activity as well, though to a lesser extent. Intriguingly, **P3**, **P7**, and **P16** share similar chemical scaffolds, suggesting they may act through a likely similar mechanism of inhibition. *In silico* predictions identified a unique cavity on Furin’s molecular surface (Pocket 3) that may easily accommodate **P3**. This exosite is located in the proximity of the catalytic site but is not a part of it. We want to highlight here that the same binding pocket was identified using two distinct approaches—blind and preferential docking—further strengthening the possibility of **P3** interaction with Furin Pocket 3. The hypothesis that **P3** docks in close proximity to the catalytic site suggests that the compound disrupts the interaction between the enzyme and the substrate, in particular amino acids distal from the scissile bond. It is worth noting at this point that the Omicron variant gained enhanced cleavability due to mutations in this specific distal area. In line with our results, permethrin, a Furin inhibitor discovered by *in silico* screening and validated against synthetic substrates, acts by targeting a likewise Pocket 3 exosite, involving Trp531 and Ala532 [44], much like the anticipated action of **P3**. The same cavity seems to be able to accommodate other possible Furin inhibitors, such as vitamin B12 and folic acid [45], naphthofluorescein [46], and epicatechin gallate [47]. Therefore, it would be compelling to explore chemical modifications of **P3** or analogous compounds.

Discovering **P3** as a novel inhibitor targeted at the enzyme Furin, specifically designed to impede the SARS-CoV-2 envelope glycoprotein maturation, marks a significant advancement in antiviral research. This breakthrough not only showcases the potential for tailored therapeutics against critical viral proteins but also underscores the importance of understanding the molecular mechanisms underlying Furin activities. Indeed, it supports the notion that Furin, as an enzyme, can be modulated by targeting regions beyond its catalytic site, known as exosites. These inhibitors, known as allosteric inhibitors, operate by binding to a site distinct from the active site, inducing conformational changes that affect enzyme activity. Recent advances have identified several small molecules that may inhibit Furin through such a mechanism, suggesting an additional layer of regulatory control that could be exploited for therapeutic purposes. The BOS-318 Furin inhibitor is particularly interesting for its allosteric function. Unlike traditional inhibitors that directly interact with the catalytic site, BOS-318 operates through a unique mechanism. It binds to a cryptic pocket near the Furin active site, which is not part of the catalytic triad. This binding induces a conformational change in Furin, specifically causing a flip in the W254 residue. This flip creates a new binding pocket that the dichlorophenyl moiety of BOS-318 fills, effectively modulating the enzyme’s activity indirectly and selectively. This allosteric mechanism allows BOS-318 to confer highly selective inhibition of Furin, which could be advantageous in therapeutic contexts where precise modulation of Furin activity is necessary without broadly affecting other proteases [31,48]. Another example of allosteric Furin inhibitors is offered by Permethrin, a recently identified compound that acts through a novel non-competitive allosteric mechanism [44]. Both BOS-318 and Permethrin provide unique perspectives on allosteric inhibition, each with a distinct interaction pattern with Furin, thus serving as useful tools in the development and analysis of new Furin inhibitors. While the well-described BOS-318 could serve as a valuable control in studies of allosteric inhibition of Furin, particularly when compared to **P3** inhibitors, Permethrin is also an attractive control. This is because Permethrin may interact with the same Furin pocket targeted by the **P3** inhibitor. The similarity in their binding sites can provide important insights into the comparative efficacy and selectivity of these inhibitors. This approach may support future studies aimed at understanding how different molecules can influence Furin function through similar or distinct allosteric mechanisms.

Considering the specificity of Furin substrates influenced by distinct amino acid sequences around the scissile bond, the allosteric inhibition could offer a targeted approach to modulate Furin activity without broadly affecting all its physiological functions. This method might allow for more selective inhibition, potentially reducing the risk of side effects associated with broader enzymatic suppression. Allosteric regulation could thus provide a nuanced control mechanism, offering benefits over traditional active-site inhibitors by potentially maintaining the enzyme physiological roles while selectively inhibiting pathological processing events. This insight is pivotal, aligning with a paradigm shift that prioritizes the inhibition of proteases without fully suppressing their enzymatic activities *in vivo*, thus avoiding potential detrimental effects.

In conclusion, by using a novel *in vitro* setting to search for Furin activity inhibitors, we screened a non-commercially available collection of small molecules (Povilas Library). Among others, we fully characterized the compound **P3**, which demonstrates promising functionality as a novel Furin inhibitor capable of selectively blocking SARS-CoV-2-derived substrate processing while leaving classical RVKR cleavage unaffected. With a molecular mass of 485.80, falling within the favorable range for drug candidates, **P3** holds promise for further drug development. In particular, the identification of **P3** as a unique chemical scaffold is particularly noteworthy as it paves the way for further enhancements in the creation of more potent and substrate-specific Furin inhibitors. Further research endeavors are essential to unlock the full therapeutic potential of **P3** and to harness its novel scaffold for the development of more effective and targeted antiviral applications.

With **P3**, we offer a potential avenue for the development of conceptually new Furin inhibitors as effective antiviral treatments to mitigate the impact of diseases like COVID-19.

## 4. Materials and Methods

### 4.1. Soluble Furin Production

sFurin consists of a soluble form of hFurin truncated before the transmembrane domain [49]. Soluble Furin (sFur) corresponds to Furin truncated before its transmembrane domain (BTMD), and the expression plasmid was kindly provided by prof. Nabil G. Seidah. The enzyme was produced by transient transfection of human embryonic kidney cells (HEK-293T) using Lipofectamine 3000 (Thermofisher, Waltham, MA, USA) and Opti-MEM medium (Gibco Fisher Scientific, Thermofisher, Waltham, MA, USA) following the manufacturer’s instructions. Cells were maintained at 37 °C, 5% CO_2_ in Advanced Dulbecco’s Modified Eagle Medium (Gibco Fisher Scientific, Thermofisher, Waltham, MA, USA) supplemented with 10% Fetal Bovine Serum Albumin (Gibco), 1% L-Glutamine, and 1% penicillin-streptomycin solution (Gibco Fisher Scientific, Thermofisher, Waltham, MA, USA, 100×). Twenty-four hours post transfection, media were collected, aliquoted (0.5 mL/aliquot), and stored at −80 °C. sFur media were used for the *in vitro* experiments without further purification [12]. sFur expression was confirmed by testing the *in vitro* activity as reported below.

### 4.2. In Vitro Assays

*In vitro* assays were performed in a 100 μL final volume, using black 96-well half-area plates (Costar). Furin substrates (Pyr-RTKR-AMC, Peptide Institute, Inc., Osaka, Japan or QTQTKSHRRAR-AMC peptide, custom made, purity ≥ 95% HPLC grade, and identity verified by mass spectrometry [MS], Genscript) or SKI-1/S1P substrate (Ac-FYISRRLL-AMC, custom made, Genscript, Piscataway, NJ, USA) concentrations were 2.5 μM, unless indicated otherwise. Reaction mixtures included 2.0 mM CaCl_2_ and were buffered at different pH values with either 25 mM sodium acetate (pH 5.5) or 25 mM 4-(2-hydroxyethyl)-1-piperazineethanesulfonic acid (HEPES; pH 6.5, 7.0, and 7.5). Each reaction contained 10 μL of sFur and sSKI-1/S1P conditioned media. The inhibitors to be tested were dissolved in dimethylsulfoxide (DMSO) and pre-incubated for 5 min with the enzyme before adding the substrate. The cleavage reactions were monitored by release of free AMC at an excitation wavelength of 360 nm and an emission wavelength of 460 nm every 3 min for 1 h at room temperature (RT) with an Infinite M200 Pro fluorescence spectrophotometer plate reader (TECAN, Mannedorf, Switzerland), unless specified otherwise. The initial rates for the hydrolysis of substrate peptide (V0) were determined by following the change in fluorescence (relative fluorescence units/min, RFU/min), plotted as a function of the substrate concentration ([S]) and fit to the Michaelis–Menten equation Vo = Vmax [S]/([S] + Km). The values of the substrate Michaelis–Menten constants (Km) and inhibitor IC50s were calculated using GraphPad Prism Version 8.0 software. All assays were performed in triplicate, and statistical analysis was carried out with GraphPad Prism Version 8.0 software.

### 4.3. High-Throughput Screening

*In vitro* high-throughput screening was established to assess the potential inhibitory effects of chemical compounds from two different libraries: the phytochemical library (Prestwick, 320 purified compounds) and the chemical fragment library (Custom made, 300 purified molecules, Povilas Kavaliauskas). All the phytochemical library compounds were dissolved in DMSO and tested at 10 μM. Molecules of the chemical fragment library were used at various concentrations in DMSO, to reach a final concentration of 20 μg/mL, unless specified otherwise (Appendix A). In total, 10 μL of sFur medium was pre-incubated with each drug for 10 min before adding the fluorogenic substrate. Tests were carried out at RT using 2.5 μM of either Pyr-RTKR-AMC or QTQTKSHRRAR-AMC—unless specified differently—in 25 mM HEPES pH 7.0 and CaCl_2_ 2.0 mM. DMSO and CMK Furin inhibitor were used as negative and positive controls, respectively. The fluorescence of cleaved AMC was measured at λ_ex_ = 360 nm, λ_em_ = 460 nm with an Infinite M200 Pro fluorescence spectrophotometer plate reader (TECAN, Mannedorf, Switzerland). Each drug was tested in duplicate. Compounds were classified as inhibitors when they were able to decrease Furin activity by at least 50% compared to DMSO treatment (considered as 100% activity). T-Tests were performed for selected inhibitory compounds to assess whether their efficacy was significant (GraphPad Prism Version 8.0).

### 4.4. General Procedure for the Preparation of Compounds **P3**, **P5**, **P7**, **P9**, **P13**, and **P16** and Their Characterization

Reagents and solvents were purchased from Sigma-Aldrich (St. Louis, MO, USA) and used without further purification. The reaction course and purity of the synthesized compounds were monitored with TLC using aluminum plates pre-coated with Silica gel at F254 nm (Merck KGaA, Darmstadt, Germany). Melting points were determined with a B-540 melting point analyzer (Büchi Corporation, New Castle, DE, USA) and were uncorrected. NMR spectra were recorded with a Brucker Avance III (400, 101 MHz) spectrometer. Chemical shifts were reported in (δ) ppm relative to tetramethylsilane (TMS) with the residual solvent as an internal reference ([D_6_]DMSO, δ = 2.50 ppm for ^1^H and δ = 39.5 ppm for ^13^C). The data are reported as follows: chemical shift, multiplicity, coupling constant [Hz], integration, and assignment. The IR spectra (ν, cm^−1^) were recorded with a Perkin–Elmer Spectrum BX FT–IR spectrometer using KBr pellets. Mass spectra were obtained with a Bruker maXis UHRTOF mass spectrometer with ESI ionization.

#### 4.4.1. Compounds **P3** and **P16**

Chemical synthesis of **P3** and **P16** is described in Figure 11 (upper panel). Briefly, a mixture of the thiazolone **1** (0.16 g, 0.39 mmol) and the corresponding aldehyde **2** (0.43 mmol) and sodium carbonate (0.22 g, 2.1 mmol) and water (3 mL) was boiled for 3 h. The resulting compounds were isolated by acidifying the cooled reaction mixture with acetic acid to pH 6. The obtained products were purified by dissolving them in 5% Na_2_CO_3_ solution (5 mL H_2_O, 0.25 g Na_2_CO_3_). The solution was filtered, and the filtrate was acidified with acetic acid to pH 6. The resulting precipitate was filtered, washed with water, and dried.

3-((5-((5-Bromothiophen-2-yl)methylene)-4-oxo-4,5-dihydrothiazol-2-yl)(3-chloro-4-methylphenyl)amino)propanoic acid (**P3**)

Yellowish solid, yield 0.16 g, 84%, m. p. 218–219 °C. IR (KBr): ν 2958 (OH); 1717, 1691 (2x C=O); 1531 (C=N) cm^−1^. ^1^H NMR (400 MHz, DMSO–d_6_) δ 2.42 (s, 3H, CH_3_), 2.64 (t, J = 7.4 Hz, 2H, CH_2_CO), 4.25 (t, J = 7.4 Hz, 2H, NCH_2_), 7.34 (d, J = 4.1 Hz, 1H, H_Ar_), 7.38 (d, J = 4.1 Hz, 1H, H_Ar_), 7.48 (dd, J = 8.2, 2.3 Hz, 1H, H_Ar_), 7.57 (d, J = 4.1 Hz, 1H, H_Ar_), 7.78 (s, 1H, H_Ar_), 7.82 (s, 1H, H_Ar_), 12.53 (br. s. 1H, OH) ppm (Appendix A). ^13^C NMR (101 MHz, DMSO–d6) δ 19.50 (CH_3_), 31.87 (CH_2_CO), 50.12 (NCH_2_), 117.00, 122.68, 127.10, 127.82, 128.49, 132.35, 133.84, 134.03, 137.87, 138.68, 140.07 (C_Ar_), 171.97, 174.97, 178.94 (C=N, 2x C=O) ppm (Appendix A). HRMS *m*/*z* calculated for C_18_H_14_BrClN_2_O_3_S_2_ [M+H]^+^: 486.9363, found: 486.9365 (Appendix A).

3-((3-Chloro-4-methylphenyl)(4-oxo-5-(thiophen-2-ylmethylene)-4,5-dihydrothiazol-2-yl)amino)propanoic acid (**P16**)

Yellowish solid, yield 0.13 g, 81%, m. p. 158–159 °C. IR (KBr): ν 2962 (OH); 1695 (2x C=O); 1522 (C=N) cm^−1^. ^1^H NMR (400 MHz, DMSO–d_6_) δ 2.42 (s, 3H, CH_3_), 2.62 (t, J = 7.4 Hz, 2H, CH_2_CO), 4.24 (t, J = 7.4 Hz, 2H, NCH_2_), 7.19 (t, J = 4.4 Hz, 1H, H_Ar_), 7.41–7.61 (m, 3H, H_Ar_), 7.72–7.82 (m, 2H, H_Ar_), 7.88 (s, 1H, H_Ar_), 12.27 (br. s. 1H, OH) ppm (Appendix A). ^13^C NMR (101 MHz, DMSO–d6) δ 19.48 (CH_3_), 32.07 (CH_2_CO), 50.06 (NCH_2_), 123.65, 127.08, 127.17, 128.03, 128.58, 128.90, 131.55, 132.33, 133.47, 133.96, 137.76, 138.34, 138.83 (C_Ar_), 172.08, 175.32, 179.19 (C=N, 2x C=O) ppm (Appendix A). HRMS *m*/*z* calculated for C_18_H_15_ClN_2_O_3_S_2_ [M+H]^+^: 407.0285, found: 407.0286 (Appendix A).

#### 4.4.2. Compound **P5**

3-((5-Chloro-2-methylphenyl)(4,9-dioxo-4,9-dihydronaphtho[2,3-d]thiazol-2-yl)amino)propanoic acid (**P5**)

A mixture of thioureido acid **3** (0.5g, 1.8 mmol), 2,3-dichloro-1,4-naphthoquinone **4** (0.49 g, 2.16 mmol), sodium acetate (2.66 g, 32.4 mmol), and acetic acid (20 mL) was heated at 80 °C for 8 h and diluted with water (30 mL). The precipitate was filtered, washed with water, dried, and recrystallized from propan-2-ol (Figure 11, middle panel).

Red solid, yield 0.51 g, 66%, m. p. 128–129 °C. IR (KBr): ν 2955 (OH); 1710 (2x C=O); 1525 (C=N) cm^−1^. ^1^H NMR (400 MHz, DMSO–d_6_) δ 2.21 (s, 3H, CH_3_), 2.79–2.86 (m, 2H, CH_2_CO), 4.10–4.21 (m, 2H, NCH_2_), 7.37–7.78 (m, 3H, H_Ar_), 7.89–7.97 (m, 2H, H_Ar_), 8.05–8.14 (m, 2H, H_Ar_), 12.41 (br. s. 1H, OH) ppm (Appendix A). HRMS *m*/*z* calculated for C_21_H_15_ClN_2_O_4_S [M+H]^+^: 427.0514, found: 427.0510 (Appendix A).

#### 4.4.3. Compound **P7** and **P9**

3-((4-Chlorophenyl)(5-(4-(dimethylamino)benzylidene)-4-oxo-4,5-dihydrothiazol-2-yl)amino)propanoic acid (**P7**)3-((4-Chlorophenyl)(4,9-dioxo-4,9-dihydronaphtho[2,3-d]thiazol-2-yl)amino)propanoic acid (**P9**)

Synthesis and characterization of **P7** and **P9** are described in [50].

#### 4.4.4. Compound **P13**

Ethyl 3-(2-nitro-6,11-dioxo-6,11-dihydro-12H-benzo[b]phenoxazin-12-yl)but-2-enoate (**P13**)

A mixture of ester **5** (0.5 g, 1.9 mmol), 2,3-dichloro-1,4-naphthoquinone **4** (0.45 g, 2 mmol), sodium carbonate (0.51 g, 4.8 mmol), and dimethyl sulfoxide (20 mL) was stirred at room temperature for 24 h and diluted with water (40 mL). The precipitate was filtered, washed with water, dried, and recrystallized from a mixture of 2-propanol and water (1:1) (Figure 11, bottom panel).

Red solid, yield 0.65 g, 81%, m. p. 210–211 °C. IR (KBr): ν 1707, 1651 (3x C=O) cm^−1^. ^1^H NMR (400 MHz, DMSO–d_6_) δ 1.05 (t, J = 7.1 Hz 3H, CH_2_CH_3_), 2.26 (s, 3H, CCH_3_); 4.02 (q, J = 7.1 Hz, 2H, CH_2_CH_3_), 6.37 (s, 1H, C=CH); 6.99 (d, J = 8.7 Hz, 1H, H_Ar_); 7.08 (s, 1H, H_Ar_); 7.71–8.03 (m, 5H, H_Ar_) ppm (Appendix A). ^13^C NMR (101 MHz, DMSO–d_6_) δ 13.82 (CH_2_CH_3_), 22.58 (CCH_3_), 60.18 (C=CH), 50.06 (NCH2), 108.16, 116.49, 119.59, 121.34, 125.16, 126.11, 129.32, 129.42, 131.37, 133.88, 134.66, 137.81, 144.45, 149.25, 151.08 (C_Ar_), 163.17, 174.83, 177.75 (3x C=O) ppm (Appendix A). HRMS *m*/*z* calculated for C_22_H_16_N_2_O_7_ [M+H]+: 421.1030, found: 421.1034 (Appendix A).

### 4.5. In Silico Analyses

*In silico* investigations were carried out on the **P3** inhibitor interacting with Furin.

**P3** inhibitor structure. The SMILES (Simplified Molecular-Input Line-Entry System) [51] identifier of the **P3** compound was derived from its IUPAC name, (3-((5-((5-bromothiophen-2-yl) methylene)-4-oxo-4,5 dihydrothiazol-2-yl)(3-chloro-4-methylphenyl)amino)propanoic acid) by OPSIN server (https://opsin.ch.cam.ac.uk/), access 14 October 2023. The three-dimensional structure of **P3** in the mol2 format was attained using Open Babel (http://www.cheminfo.org/Chemistry/Cheminformatics/FormatConverter/index.html, access 14 October 2023) [52]. Finally, the obtained structure of **P3** underwent optimization for the MM2 force field using Chem3D 21.0.0, a module integrated within ChemOffice 21.0.0.

#### 4.5.1. Furin Enzyme Structure

The three-dimensional structure of Furin was obtained from the Protein Data Bank (PDB) database (https://www.rcsb.org/, access on 1 October 2023) [53]. Currently, there are forty registered structures for human Furin, encompassing both X-ray crystallography and NMR-derived structures. Here, we used the crystal structure of unglycosylated apo human Furin (PDB id 4Z2A), with a resolution of 1.89 Å. Subsequently, we performed protein structure optimization, using Swiss-PDB Viewer v4.1.0.

#### 4.5.2. Molecular Docking

We employed two different approaches. The first approach was blind docking of **P3** on Furin. To this purpose, we used the Achilles Blind Docking Server (https://bio-hpc.ucam.edu/achilles/, access on 1 October 2023), which works based on a customized version of AutoDock Vina. The second approach used preferential docking of **P3** into various predicted Furin pockets (exosites). FTPocketWeb 1.0.1 online tool (https://durrantlab.pitt.edu/fpocketweb/, default parameters, access on 1 October 2023) [40] was used to predict the pockets on the Furin surface. The server provided a druggability score for each predicted pocket. The three highest-scoring predicted pockets were selected for further analyses. AutoDockTools-1.5.7 [41] was used to prepare the protein for molecule docking. Water molecules and ligands due to crystallization conditions (identified by HETAM records—hetero atoms—in the PDB file) were removed. Polar hydrogens were added, Kolman charges were incorporated, and non-polar hydrogens were merged, exporting the final model in the PDBQT file format. The establishment of the grid box was based on the spatial coordinates of the predicted pockets. Docking operations were performed using AutoDock 4.2.6 [41] and AutoDock Vina v1.2.5 [42,43]. Finally, docking results were analyzed and visualized by BIOVIA Discovery Studio Visualizer 2021 Client (https://discover.3ds.com/discovery-studio-visualizer-download, access on 1 October 2023) and PyMOL.

Further *in silico* studies were performed to check similarities between the hits emerging from the *in vitro* screening using RDKit package ver. 2023.09.1 (Open-source cheminformatics. https://www.rdkit.org, access on 1 October 2023). **P3** worked as the reference model. Similarity maps were generated based on the Morgan fingerprint and Tanimoto Metric (https://www.rdkit.org/docs/GettingStartedInPython.html#fingerprinting-and-molecular-similarity, access on 1 October 2023).

### 4.6. Sustainability

Sustainability lies at the heart of our operations, guiding our every decision and action. We understand the significance of responsibly managing the resources involved in our work. By meticulously tracking and acknowledging the resources utilized, we pave the way for more informed and strategic planning of future experiments. This conscientious approach not only aligns with our values but also ensures that we contribute positively to the environment and the community around us.

In our research, conscious efforts were made to minimize plastic usage through the adoption of sustainable laboratory practices, such as utilizing reusable glassware and eco-friendly alternatives wherever feasible, thereby contributing to reduce environmental impact in our scientific endeavors. In particular, special attention was given to the screening step that represented the main source of plastic use. Thus, to mitigate environmental impact, every 96-well plate used in our experiments underwent a systematic recycling process in accordance with well-established internal protocols, ensuring their reuse and promoting sustainability in our research practices.

Based on the summary provided in our “Green Book” (GB), we found that we needed approximately 12 kg of plastics, of which only a fraction (5%) could be recycled. The remaining portion was categorized as hazardous materials according to Italian regulations, necessitating specific treatment procedures (Appendix A). We advocate for heightened awareness among researchers regarding the disposable items regularly utilized in their laboratory settings. Implementing a straightforward diary, such as the Green Book (GB), to record rough estimates of plastic usage can prove invaluable in identifying areas for improvement and implementing greener alternatives.

## Figures and Tables

**Figure 1 ijms-25-05079-f001:**
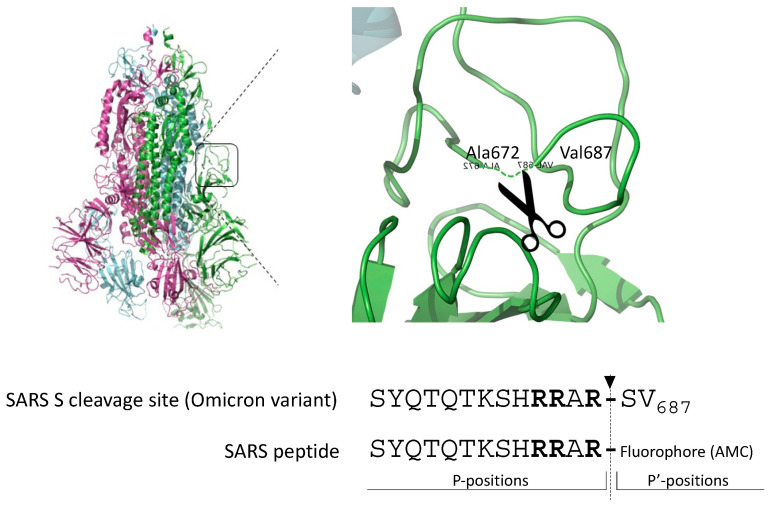
Cleavage site of the envelope glycoprotein of SARS-CoV-2. Schematic representation of the engineered peptide SYQTQTKSHRRAR-AMC (SARS peptide) designed for assessing Furin activity *in vitro*. The SARS-CoV2 peptide is derived from the scissile bond between S1/S2 of the envelope glycoprotein S of SARS-CoV-2, featuring the crystal structure illustrated in the figure. Arrow indicates the position of Furin-mediated cleavage; in bold, key basic residues recognized by Furin.

**Figure 2 ijms-25-05079-f002:**
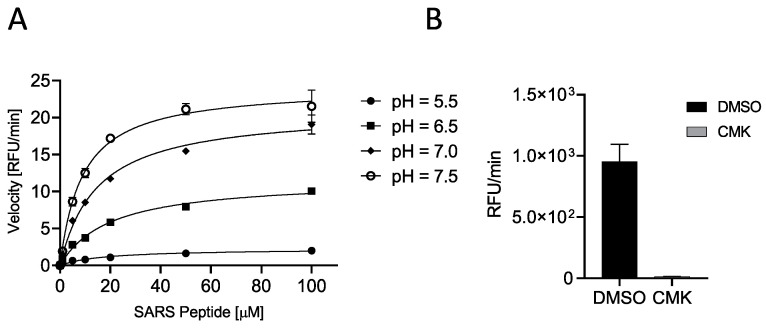
Characterization of SARS peptide (**A**) SYQTQTKSHRRAR-AMC (SARS-CoV2 peptide) cleavability was assessed with soluble human Furin (sFur) under varying pH conditions (5.5, 6.5, 7.0, and 7.5) and peptide concentrations (0.01–100 μM) by monitoring fluorescence (λ_ex_ = 360 nm/λ_em_ = 460 nm) released over time. Reaction velocities were calculated and plotted vs. substrate concentrations. Measurements indicated concentration-dependent cleavage, particularly pronounced under neutral/slightly basic pH conditions. (**B**) Validation of Furin-dependent processing of SARS-CoV2 peptide using the Furin-specific Decanoyl-RVKR-chloromethylketone (CMK) inhibitor. sFur was preincubated with 0.1 μM CMK prior to adding 5 μM fluorogenic substrate. Z was calculated using the following formula proposed by Zhang and colleagues [39]: Z = 1 − [(mean of positive controls − mean of negative controls) 3 × (standard deviation of positive controls + standard deviation of negative controls)] Z score is 0.65, affirming the SARS-CoV2 peptide suitability for high-throughput screening. RFU: Relative Fluorescence Unit.

**Figure 3 ijms-25-05079-f003:**
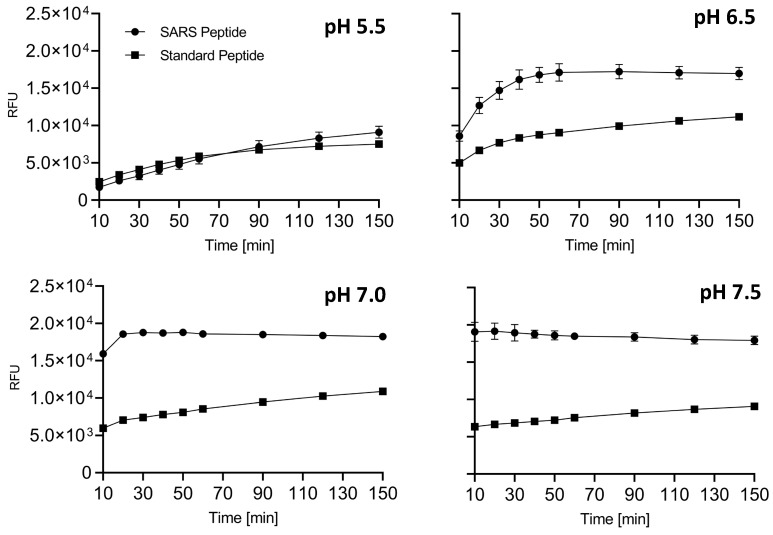
Side-by-side comparison of SARS and standard peptide substrates. SARS or standard peptides (5 μM) were incubated with 20 μL of freshly prepared sFur buffered at various pH values (5.5, 6.5, 7.0, and 7.5). Cleavage kinetics were monitored by recording fluorescence (λ_ex_ = 360 nm/λ_em_ = 460 nm) over time. The SARS substrate consistently outperforms the standard one, showcasing its suitability for studying Furin protease activity. The fluorescence intensity immediately after enzyme addition is notably higher for the SARS peptide at pH > 6.5, aligning with the kinetic characteristics observed for the entire envelope. RFU: Relative Fluorescence Unit.

**Figure 4 ijms-25-05079-f004:**
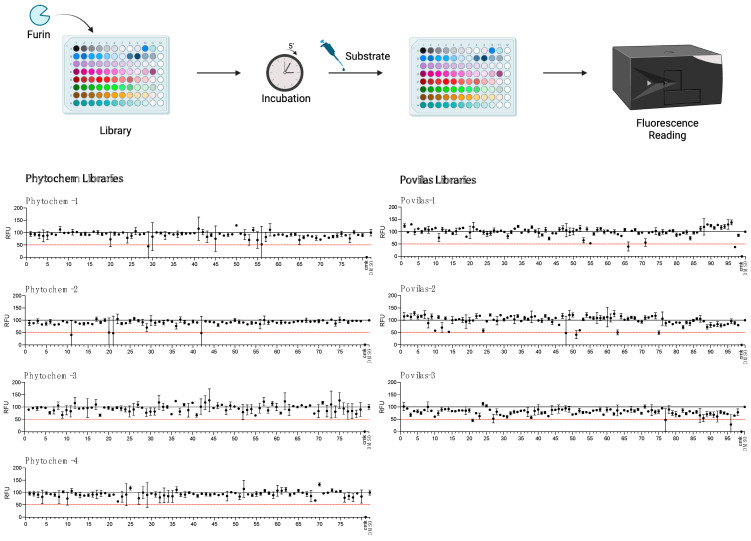
HTS for Furin inhibitors using the SARS-CoV2 substrate. Two libraries (Prestwick Phytochemical and Povilas small-compound collections) were screened for their ability to block Furin enzymatic activity. After pre-incubation of 20 μL of freshly prepared sFur with the various compounds (see the Materials and Methods for details), 5 µM of SARS substrate was added, and the fluorescence was recorded (λ_ex_ = 360 nm/λ_em_ = 460 nm) for 1 h. CMK and DMSO were used as positive and negative controls, respectively. The reported data represent the velocity rate (RFU/min) of the reaction normalized to DMSO, arbitrarily set to 100%. Each point is the mean of two independent sets of experiments. DMSO: Dimethylsulphoxide; RFU: Relative Fluorescence Unit.

**Figure 5 ijms-25-05079-f005:**
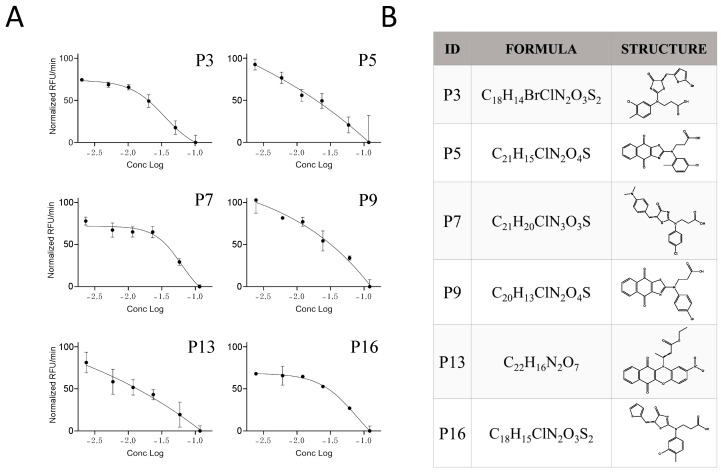
Potency of the inhibitor hits. (**A**) Dose-dependent inhibition of Furin by selected inhibitors towards SARS-CoV2 peptide processing was achieved by pre-incubation of 20 μL of freshly prepared sFur with the indicated compounds (**P3**, **P5**, **P7**, **P9**, **P13**, and **P16**) in the 100–2.5 μg/mL range prior to adding 2.5 μM SARS-CoV2 peptide substrate. Fluorescence was then recorded over time for 1 h. Each point is the mean of three independent experiments, and it represents the velocity rate (RFU/min) of the processing reaction. (**B**) Chemical formula and chemical structure of **P3**, **P5**, **P7**, **P9**, **P13**, and **P16**. RFU: Relative Fluorescence Unit.

**Figure 6 ijms-25-05079-f006:**
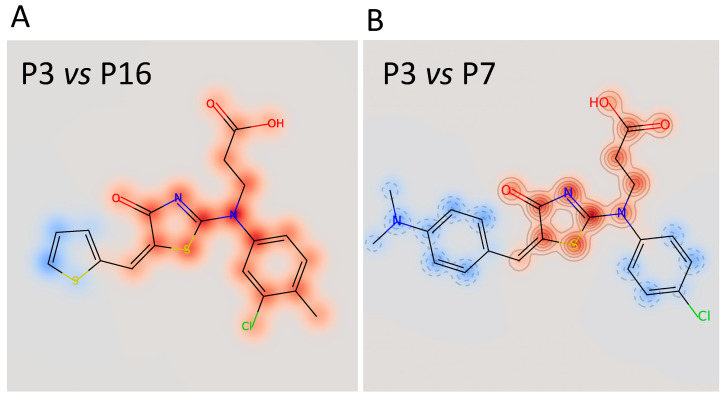
Superimposition of the chemical structure of **P3**/**P16** (**A**) and **P3**/**P7** (**B**). The red halo indicates high similarity; the blue halo indicates low similarity.

**Figure 7 ijms-25-05079-f007:**
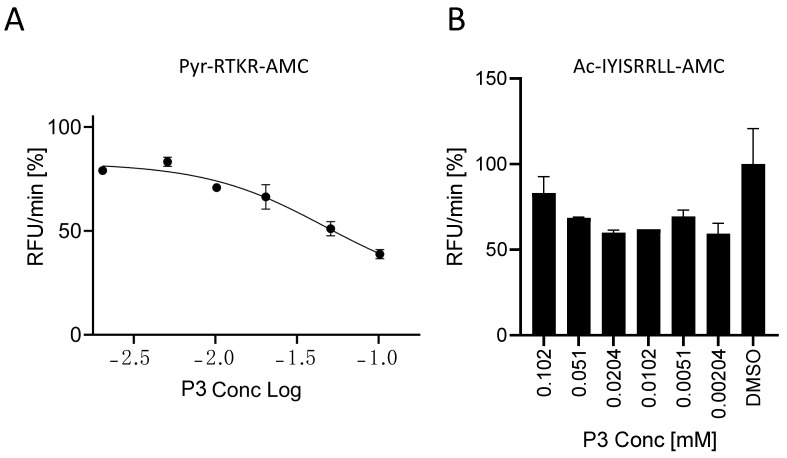
Characterization of the **P3** inhibitor. In all, 20 μL of freshly prepared sFur was incubated with increasing concentrations of **P3** prior to adding the Pyr-RTKR-AMC substrate (**A**) or Ac-IYISRRLL-AMC (**B**). Fluorescence was then recorded over time for 1 h. Each point is the mean of three independent experiments, and it represents the velocity rate (RFU/min) of the processing reaction.

**Figure 8 ijms-25-05079-f008:**
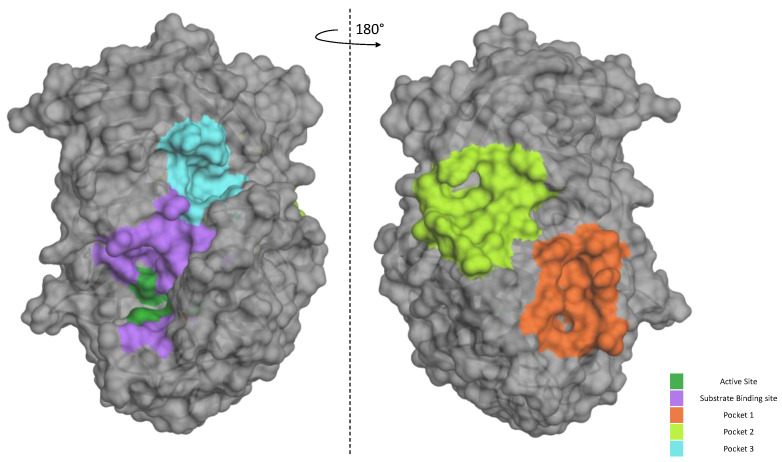
Furin exosites. The structure of unglycosylated apo human Furin (PDB ID, 4Z2A) was used to predict grooves on the Furin surface structure. FTPocketWeb 1.0.1 online tool (https://durrantlab.pitt.edu/fpocketweb/, default parameters, access on 14 October 2023) was used to identify three major pockets: Pocket 1 (orange), Pocket 2 (light green), and Pocket 3 (cyan). The catalytic pocket is depicted in violet, whereas the dark green color identifies the catalytic triad.

**Figure 9 ijms-25-05079-f009:**
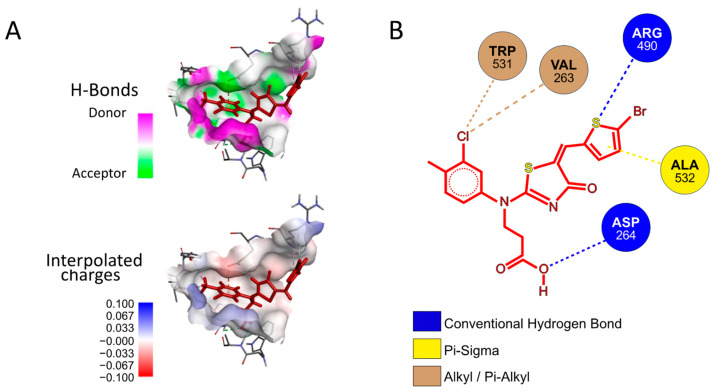
Docking of **P3** into Furin Pocket 3. (**A**) Low-energy binding conformations of the **P3** ligand and Furin complexes generated by AutoDock VINA. (**B**) Ligplot of the **P3** pose at the Furin Pocket 3.

**Figure 10 ijms-25-05079-f010:**
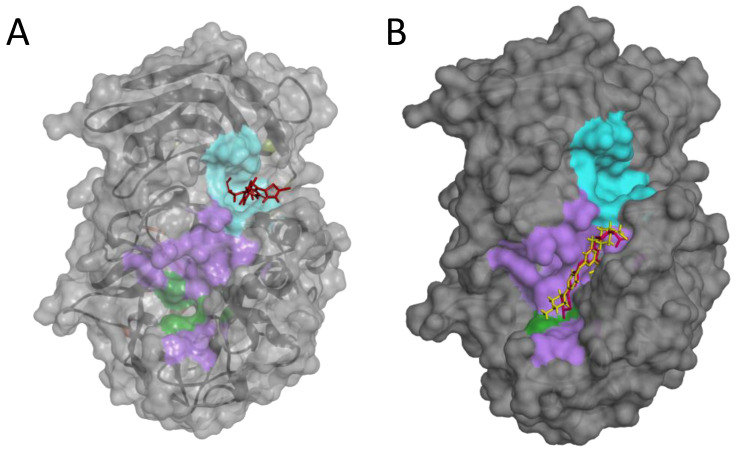
Blind docking of **P3** or BOS-318 on the Furin structure. Blind docking of **P3** (**A**) or the Furin inhibitor BOS-318 (**B**) to the surface of the unglycosylated apo human Furin (PDB ID, 4Z2A) using the Achilles Blind Docking Server (https://bio-hpc.ucam.edu/achilles/, access 14 October 2023). The violet indicates the catalytic pocket; the catalytic triad is in green; the cyan indicates the exosite where BOS-318 interacts with Furin.

**Figure 11 ijms-25-05079-f011:**
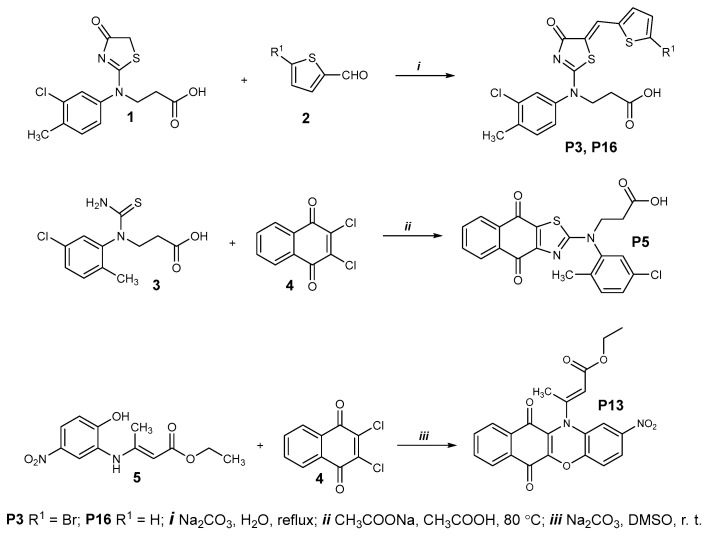
Chemical synthesis of **P3**, **P5**, **P13**, and **P16**.

**Table 1 ijms-25-05079-t001:** SARS peptide substrate kinetic digestion values.

pH	Vmax RFU/min	Km [μM]
5.5	2.27	17.54
6.5	11.69	19.90
7.0	21.04	14.70
7.5	24.2	8.94

## Data Availability

The original contributions presented in the study are included in the article/Appendix A, further inquiries can be directed to the corresponding author/s.

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
