# Peer review of "Screening of Small-Molecule Libraries Using SARS-CoV-2-Derived Sequences Identifies Novel Furin Inhibitors"

_ijms, 2024, doi:10.3390/ijms25105079_

Round 1

Reviewer 1 Report

Comments and Suggestions for Authors

In the presented manuscript, Pasquato et al describe screening of small molecule libraries using SARS-CoV-2 derived sequences for the identification of novel Furin inhibitors. The authors engineered peptide sequence that serves as an alternative Furin substrate for the inhibition tests in comparison to the one employed usually. Inhibition was investigated by employing commercially available and custom made libraries of small organic molecules. Although commercially available libraries did not have any significant hits, in the custom-made library several compounds were identified as potential inhibitors. Inhibition process by the most potent inhibitor P3 was further investigated by docking simulations.

Overall, the manuscript is well presented and easy to follow. However, there is insufficient experimental data that support identity of compounds, and hence conclusions drawn. Compounds P3, P5, P7, P9, P13 and P16 need to be fully characterized (1H and 13C NMR, IR, HRMS, including images of NMR spectra) if they are new compounds, or their previous characterization should be referenced, in which case 1H NMR shifts and spectra images would suffice. In addition, some sort of structural characterization of engineered peptide should also be present, or referenced to, at least HRMS and data regarding purity.

Minor: Chapter 2.5 should be removed from the manuscript, as it does contribute to the discussion. It can be placed in the SI. Introduction should include more information about this engineered peptide sequence if it was utilized before, or make clear that this is the first employment of this sequence.

Author Response

Answer to Reviewer 1

Reviewer 1: In the presented manuscript, Pasquato et al describe screening of small molecule libraries using SARS-CoV-2 derived sequences for the identification of novel Furin inhibitors. The authors engineered peptide sequence that serves as an alternative Furin substrate for the inhibition tests in comparison to the one employed usually. Inhibition was investigated by employing commercially available and custom made libraries of small organic molecules. Although commercially available libraries did not have any significant hits, in the custom-made library several compounds were identified as potential inhibitors. Inhibition process by the most potent inhibitor P3 was further investigated by docking simulations. Overall, the manuscript is well presented and easy to follow.

Answer: We thank you for the constructive comments and suggestions. We appreciate your feedback on the clarity and presentation of our manuscript.

Reviewer 1: However, there is insufficient experimental data that support identity of compounds, and hence conclusions drawn. Compounds P3, P5, P7, P9, P13 and P16 need to be fully characterized (1H and 13C NMR, IR, HRMS, including images of NMR spectra) if they are new compounds, or their previous characterization should be referenced, in which case 1H NMR shifts and spectra images would suffice.

Answer: We have carefully considered the comments regarding the need for more comprehensive experimental data to support the identity of the compounds discussed in our study. We are pleased to inform you that we have performed additional NMR, IR and mass spectrometry analyses, and the revised manuscript includes these data to substantiate the identity and characterization of the compounds.

Characterization of Compounds P3, P5, P7, P9, P13, and P16:

In response to your suggestion, we have included both 1H and 13C NMR, as well as IR and MS data, for compounds P3, P16, and P13. Additionally, 1H NMR, IR and MS data have also been described for compound P5.A summary summarizing ^1H NMR chemical shifts for each referenced compound to allow for easy comparison.

Images of the NMR and MS spectra for these molecules have been added to the supplementary information section, providing clear evidence of their structures. The synthesis and characterization of compounds P7 and P9 has been previously reported, and reference to these studies is now available in the revised manuscript (Ref. # 49 in the revised manuscript, Anusevičius, K.; Jonuškienė, I.; Mickevičius, V. Synthesis and antimicrobial activity of N-(4-chlorophenyl)-balanine derivatives with an azole moiety. Monatshefte fuer Chemie - Chemical Monthly 2013, 144, 1883−1891, doi:10.1007/s00706-013-1074-8.)

The following text including Fig. 11 and supplementary Fig. S1-S11 has been added (lines 545-625 of the revised manuscript):

5.4 General procedure for the preparation of compounds P3, P5, P7, P9, P13, and P16 and their characterization

Reagents and solvents were purchased from Sigma-Aldrich (St. Louis, MO, USA) and used without further purification. The reaction course and purity of the synthesized compounds were monitored by TLC using aluminium plates pre-coated with Silica gel with F254 nm (Merck KGaA, Darmstadt, Germany). Melting points were determined with a B-540 melting point analyser (Büchi Corporation, New Castle, DE, USA) and are uncorrected. NMR spectra were recorded on a Brucker Avance III (400, 101 MHz) spectrometer. Chemical shifts were reported in (δ) ppm relative to tetramethylsilane (TMS) with the residual solvent as internal reference ([D6]DMSO, δ = 2.50 ppm for 1H and δ = 39.5 ppm for 13C). Data are reported as follows: chemical shift, multiplicity, coupling constant [Hz], integration and assignment. IR spectra (ν, cm-1) were recorded on a Perkin–Elmer Spectrum BX FT–IR spectrometer using KBr pellets. Mass spectra were obtained on Bruker maXis UHRTOF mass spectrometer with ESI ionization.

5.4.1 Compounds P3 and P16

Chemical synthesis of P3 and P16 is described in Figure 11 (upper panel). Briefly, a mixture of the thiazolone 1 (0.39 mmol) and the corresponding aldehyde 2 (0.43 mmol) and sodium carbonate (0.22 g, 2.1 mmol) and water (3 mL) was boiled for 3 h. The resulting compounds are isolated by acidifying the cooled reaction mixture with acetic acid to pH 6. The obtained products are purified by dissolved them in 5% Na2CO3solution (5 ml H2O, 0.25 g Na2CO3). The solution is filtered off, the filtrate is acidified with acetic acid to pH 6. The resulting precipitate is filtered, washed with water and dried.

Figure 11. Chemical synthesis of P3, P5, P13 and P16

3-((5-((5-Bromothiophen-2-yl)methylene)-4-oxo-4,5-dihydrothiazol-2-yl)(3-chloro-4-methylphenyl)amino)propanoic acid (P3)

Yellowish solid, yield 0.16 g, 84%, m. p. 218–219 °C. IR (KBr): ν 2958 (OH); 1717, 1691 (2x C=O); 1531 (C=N) cm–1. 1H NMR (400 MHz, DMSO–d6) δ 2.42 (s, 3H, CH3), 2.64 (t, J = 7.4 Hz, 2H, CH2CO), 4.25 (t, J = 7.4 Hz,2H, NCH2), 7.34 (d, J = 4.1 Hz, 1H, HAr), 7.38 (d, J = 4.1 Hz, 1H, HAr), 7.48 (dd, J = 8.2, 2.3 Hz, 1H, HAr), 7.57 (d, J = 4.1 Hz, 1H, HAr), 7.78 (s, 1H, HAr), 7.82 (s, 1H, HAr), 12.53 (br. s. 1H, OH) ppm (Fig. S1). 13C NMR (101 MHz, DMSO–d6) δ 19.50 (CH3), 31.87 (CH2CO), 50.12 (NCH2), 117.00, 122.68, 127.10, 127.82, 128.49, 132.35, 133.84, 134.03, 137.87, 138.68, 140.07 (CAr), 171.97, 174.97, 178.94 (C=N, 2x C=O) ppm (Fig. S2). HRMS m/z calculated for C18H14BrClN2O3S2 [M+H]+: 486.9363, found: 486.9365 (Fig. DS3).

3-((3-Chloro-4-methylphenyl)(4-oxo-5-(thiophen-2-ylmethylene)-4,5-dihydrothiazol-2-yl)amino)propanoic acid (P16)

Yellowish solid, yield 0.13 g, 81%, m. p. 158–159 °C. IR (KBr): ν 2962 (OH); 1695 (2x C=O); 1522 (C=N) cm–1. 1H NMR (400 MHz, DMSO–d6) δ 2.42 (s, 3H, CH3), 2.62 (t, J = 7.4 Hz, 2H, CH2CO), 4.24 (t, J = 7.4 Hz, 2H, NCH2), 7.19 (t, J = 4.4 Hz, 1H, HAr), 7.41−7.61 (m, 3H, HAr), 7.72−7.82 (m, 2H, HAr), 7.88 (s, 1H, HAr), 12.27 (br. s. 1H, OH) ppm (Fig. S4). 13C NMR (101 MHz, DMSO–d6) δ 19.48 (CH3), 32.07 (CH2CO), 50.06 (NCH2), 123.65, 127.08, 127.17, 128.03, 128.58, 128.90, 131.55, 132.33, 133.47, 133.96, 137.76, 138.34, 138.83 (CAr), 172.08, 175.32, 179.19 (C=N, 2x C=O) ppm (Fig. S5). HRMS m/z calculated for C18H15ClN2O3S2 [M+H]+: 407.0285, found: 407.0286 (Fig. S6).

5.4.2 Compound P5

3-((5-Chloro-2-methylphenyl)(4,9-dioxo-4,9-dihydronaphtho[2,3-d]thiazol-2-yl)amino)propanoic acid (P5)

A mixture of thioureido acid 3 (0.5g, 1.8 mmol), 2,3-dichloro-1,4-naphthoquinone 4 (0.49g, 2.16 mmol), sodium acetate (2.66 g, 32.4 mmol), and acetic acid (20 mL) was heated at 80 °C for 8 h, and diluted with water (30 mL). The precipitate was filtered off, washed with water, dried, and recrystallized from propan-2-ol (Fig. 11, middle panel).

Red solid, yield 0.51 g, 66%, m. p. 128–129 °C. IR (KBr): ν 2955 (OH); 1710 (2x C=O); 1525 (C=N) cm–1. 1H NMR (400 MHz, DMSO–d6) δ 2.21 (s, 3H, CH3), 2.79−2.86 (m, 2H, CH2CO), 4.10−4.21 (m, 2H, NCH2), 7.37−7.78 (m, 3H, HAr), 7.89−7.97 (m, 2H, HAr), 8.05−8.14 (m, 2H, HAr), 12.41 (br. s. 1H, OH) ppm (Fig. S7). HRMS m/z calculated for C21H15ClN2O4S [M+H]+: 427.0514, found: 427.0510 (Fig.S8).

5.4.3 Compound P7 and P9

3-((4-Chlorophenyl)(5-(4-(dimethylamino)benzylidene)-4-oxo-4,5-dihydrothiazol-2-yl)amino)propanoic acid (P7)

3-((4-Chlorophenyl)(4,9-dioxo-4,9-dihydronaphtho[2,3-d]thiazol-2-yl)amino)propanoic acid (P9)

Synthesis and characterization of P7  and P9 are described in [49].

5.4.4 Compound P13

Ethyl 3-(2-nitro-6,11-dioxo-6,11-dihydro-12H-benzo[b]phenoxazin-12-yl)but-2-enoate (P13)

A mixture of ester 5 (0.5g, 1.9 mmol), 2,3-dichloro-1,4-naphthoquinone 4 (0.45g, 2 mmol), sodium carbonate (0.51 g, 4.8 mmol), and dimethyl sulfoxide (20 mL) was stireed at room temperature for 24 h, and diluted with water (40 mL). The precipitate was filtered off, washed with water, dried, and recrystallized from a mixture of 2-propanol and water (1:1) (Fig. 11, bottom panel).

Red solid, yield 0.65 g, 81%, m. p. 210–211 °C. IR (KBr): ν  1707, 1651 (3x C=O) cm–1. 1H NMR (400 MHz, DMSO–d6) δ 1.05 (t, J = 7.1 Hz 3H, CH2CH3), 2.26 (s, 3H, CCH3); 4.02 (q, J = 7.1 Hz, 2H, CH2CH3), 6.37 (s, 1H, C=CH); 6.99 (d, J = 8.7 Hz, 1H, HAr); 7.08 (s, 1H, HAr); 7.71−8.03 (m, 5H, HAr) ppm (Fig. S9). 13C NMR (101 MHz, DMSO–d6) δ 13.82 (CH2CH3), 22.58 (CCH3), 60.18 (C=CH), 50.06 (NCH2), 108.16, 116.49, 119.59, 121.34, 125.16, 126.11, 129.32, 129.42, 131.37, 133.88, 134.66, 137.81, 144.45, 149.25, 151.08 (CAr), 163.17, 174.83, 177.75 (3x C=O) ppm (Fig. S10). HRMS m/z calculated for C22H16N2O7 [M+H]+: 421.1030, found: 421.1034 (Fig. S11).

Reviewer 1: In addition, some sort of structural characterization of engineered peptide should also be present, or referenced to, at least HRMS and data regarding purity.

Answer: The SARS peptide was purchased from the GenScript company. Although GenScript provided evidence of the peptide's purity and identity, these data were not generated by our team and therefore were not included in the main text of our manuscript. However, in response to your comments, we have added a statement in the Materials and Methods section indicating that the peptide was supplied with a purity greater than 95% and that its identity was verified by the company.

Old version, lines 479-480

Furin substrates (Pyr-RTKR-AMC, Peptide Institute, Inc. or QTQTKSHRRAR-AMC peptide, custom made Genscript)

Revised manuscript, lines 506-508:

Furin substrates (Pyr-RTKR-AMC, Peptide Institute, Inc. or QTQTKSHRRAR-AMC peptide, custom made, purity³95% HPLC grade, identity verified by mass spectrometry [MS], Genscript).

Reviewer 1: Minor: Chapter 2.5 should be removed from the manuscript, as it does contribute to the discussion. It can be placed in the SI.

Answer: We have removed Chapter 2.5 as suggested, acknowledging that it did not contribute directly to the core discussion. This section has now been relocated to the Materials and Methods section (5.5) and the attached table was included in the Supplementary Information (SI) to maintain its accessibility for interested readers.

Reviewer 1: Introduction should include more information about this engineered peptide sequence if it was utilized before, or make clear that this is the first employment of this sequence.

Answer: We have revised the introduction to addresses the reviewer’s suggestion to elucidate the novelty and prior application status of the peptide. It now clearly states that this sequence is derived from Omicron SARS-CoV-2 envelope glycoprotein and it represents the first utilization of its kind.

Old version, lines 128-138

Keeping this in mind, we have designed an assay for the in vitro screening of novel Furin inhibitors specifically targeting the cleavage site of the envelope glycoprotein of SARS-CoV-2, using the SYQTQTKSHRRAR-(7-Amido-4-methylcoumarin) [AMC] fluorogenic peptide. Standard in vitro Furin activity is assessed with the Pyroglutamic(Pyr)-RTKR-AMC peptide [36], which is much shorter and bears a different amino acid sequence. Pyr-RTKR-AMC is the gold standard substrate used for Furin inhibitor screening, e.g. [37,38]. Here, we characterized a new SARS-CoV-2 derived Furin substrate that is suitable for high throughput screening (HTS). Proof-of-concepts assays using commercially available and custom-made compounds libraries identified a novel Furin inhibitor which can block the processing of the viral substrate more efficiently than that of Pyr-RTKR-AMC.

Introduction should include more information about this engineered peptide sequence if it was utilized before, or make clear that this is the first employment of this sequence.

Revised manuscript, lines 128-143:

Considering this context, our team has developed an assay to facilitate the in vitro screening of novel Furin inhibitors that target the cleavage site of the SARS-CoV-2 envelope glycoprotein, utilizing a uniquely designed fluorogenic peptide, SYQTQTKSHRRAR-(7-Amido-4-methylcoumarin) [AMC]. This peptide, an innovative construct proposed by our research group and mimicking the Omicron variant cleavage site of SARS-CoV-2 envelope glycoprotein, represents the first use of this sequence in such applications. Indeed, typical in vitro Furin activity assays employ the Pyroglutamic(Pyr)-RTKR-AMC peptide [36], which is much shorter and bears a different amino acid sequence. As a matter of fact, Pyr-RTKR-AMC is the gold standard reference used for Furin inhibitor screening, e.g. [37,38]. Here, we successfully replaced Pyr-RTKR-AMC with the new SARS-CoV-2 derived Furin substrate SYQTQTKSHRRAR-AMC, showing that the latter is suitable for Furin high throughput screening (HTS). Furthermore, proof-of-concepts assays using commercially available and custom-made compounds libraries identified a novel Furin inhibitor which can block the processing of the viral de-rived substrate more efficiently than that of Pyr-RTKR-AMC.

Reviewer 2 Report

Comments and Suggestions for Authors

In this study by Jorkesh et. al. the authors have developed a florescence based high throughout assay using an Omicron variant derived (modified) peptide for furin protein. The authors also screened small molecule libraries to select compounds that inhibit furin activity using the developed method. Authors Additionally, the binding site of the selected compound (scaffold) is predicted using in silico methods. I would like to applaud authors for their sustainability initiative in science. Overall, the manuscript is well written and clearly describes their effort in a way which can be reproduced by other researchers. My major concerns lie with the computational aspect of the work, and I feel more needs to be done to convince that the small molecule binds to the site that authors have predicted. There are some other minor concerns too.

The authors use Autodock and Autodock vina (most likely rigid docking) to dock these compounds along with blind docking. The docking algorithms do not sufficiently account for effect of time, dynamics, and solvation on the protein-small molecule complex. I believe a short molecular dynamics (few 100ns) run should be employed to show that the compound is stable in its docked orientation and would not diffuse back in the solution. This problem is serious in this particular case as the binding affinity of the complex is shown to be of the order of ~-7 kcal/mol (Autodock and vina). -7 kcal/mol from Autodock based docking tools is not a significant binding affinity and from my last 10 years of experience many compounds would give such a binding affinity in a docking screen (last week we docked ~80 million small molecules to a clinically relevant target and mode binding affinity lied between -7 and -8 kcal/mol (unpublished results)). Docking should therefore be employed only as a screening tool with more validation performed based on docking predictions.

Also, the authors might be indicating to an allosteric control, I understand investigating it is beyond scope of this manuscript but discussing the possibility and its implications can add depth to the discussion section.

Authors should address more about the effect of mutations to the SARS-CoV2 furin substrate residues as indicated in previous literature in the introduction, this would provide important context and enhance the understanding of the study's findings.

I would like the authors to comment on the publication: https://doi.org/10.1371/journal.ppat.1009246

Line 226-227 indicates towards Table 4 but the manuscript seems to have only 2 Tables.

Author Response

Answer to Reviewer 2

Reviewer 2: In this study by Jorkesh et. al. the authors have developed a florescence based high throughout assay using an Omicron variant derived (modified) peptide for furin protein. The authors also screened small molecule libraries to select compounds that inhibit furin activity using the developed method. Authors Additionally, the binding site of the selected compound (scaffold) is predicted using in silico methods. I would like to applaud authors for their sustainability initiative in science. Overall, the manuscript is well written and clearly describes their effort in a way which can be reproduced by other researchers. My major concerns lie with the computational aspect of the work, and I feel more needs to be done to convince that the small molecule binds to the site that authors have predicted. There are some other minor concerns too.

The authors use Autodock and Autodock vina (most likely rigid docking) to dock these compounds along with blind docking. The docking algorithms do not sufficiently account for effect of time, dynamics, and solvation on the protein-small molecule complex. I believe a short molecular dynamics (few 100ns) run should be employed to show that the compound is stable in its docked orientation and would not diffuse back in the solution. This problem is serious in this particular case as the binding affinity of the complex is shown to be of the order of ~-7 kcal/mol (Autodock and vina). -7 kcal/mol from Autodock based docking tools is not a significant binding affinity and from my last 10 years of experience many compounds would give such a binding affinity in a docking screen (last week we docked ~80 million small molecules to a clinically relevant target and mode binding affinity lied between -7 and -8 kcal/mol (unpublished results)). Docking should therefore be employed only as a screening tool with more validation performed based on docking predictions.

Answer: We thank the reviewer for highlighting a critical point regarding the limitations of Autodock and Autodock Vina in our docking simulations. While the primary focus of this study was to establish a new in vitro screening assay framework against Furin, the comprehensive characterization of novel Furin inhibitors—including a detailed analysis of the P3 mechanism of action supported by molecular dynamics of Furin-P3 complexes—will be addressed in future experiments as we continue to develop this project. We acknowledge the necessity for a more comprehensive analysis that encompasses dynamics and solvation effects on the protein-small molecule complex. Indeed, molecular docking provides a snapshot of the interaction, capturing only a single moment. In contrast, molecular dynamics simulation tracks the complex over time, offering a deeper understanding of the interaction dynamics between the two partners. However, due to limited computational resources and time, we could not extend our molecular dynamics simulations beyond 50 nanoseconds. Stability of the Furin backbone and P3 – Furin (4Z2A) complex was evaluated using molecular dynamic (MD) trajectories. Root mean square deviation (RMSD) profiles achieved from the MD simulations are illustrated in Fig. R1. Average RMSD values with standard deviations of Furin backbone and P3 – Furin complex were 0.12 ± 0.016 nm, 0.17 ± 0.042 nm, respectively. P3 with Furin complex displayed a large fluctuation up to 35 ns and was slowly stabilized around 1 nm at the end of simulation.

Figure R1. Furin backbone and P3 – Furin (4Z2A) complex molecular dynamic

P3 - Furin (4Z2A)

page1image601042928 page1image601043136

Additional parameter, root mean square fluctuation (RMSF), was further investigated for P3 – Furin complex. The RMSF plot demonstrated that the interacting residues including Val263, Asp264, Arg490, Trp531, and Ala532 (pocket 3) had the average RMSF 0.057 ± 0.012 nm and average RMSF for residues in the substrate binding site including Asp154, Asp191, Asn192, Glu236, Ser253, Trp254, Gly255, Pro256, Gly257, Asp258, Asp264, Ala292, Ser293, Gly294, Asn295, Asp306, Tyr308, and Ser368 had 0.098 ± 0.035 nm.

These preliminary data provide only an initial glimpse into the interaction between Furin and P3. Extended simulations are essential for a more comprehensive understanding, and such longer dynamics are planned as part of our ongoing experiments for the next publication. To underscore the significance of molecular dynamics and to provide a balanced interpretation of molecular docking as a useful but limited tool, we have included the following statement in the main text:

Revised manuscript, lines 351-355

While molecular docking provides a static snapshot of the interaction between the enzyme and the inhibitor, molecular dynamics offer a more comprehensive view, revealing continuous interaction details over time. Accordingly, in the forthcoming experiments we plan to use molecular dynamics to gain a better understanding of the mechanism of action of this inhibitor.

Reviewer 2: Also, the authors might be indicating to an allosteric control, I understand investigating it is beyond scope of this manuscript but discussing the possibility and its implications can add depth to the discussion section.
Answer: The reviewer 2 suggestion regarding the discussion of allosteric control in the manuscript has been thoroughly addressed.
Revised manuscript, lines 442-475:
Recent advances have identified several small molecules that may inhibit Furin through such a mechanism, suggesting an additional layer of regulatory control that could be exploited for therapeutic purposes. The BOS-318 Furin inhibitor is particularly interesting for its allosteric function. Unlike traditional inhibitors that directly interact with the catalytic site, BOS-318 operates through a unique mechanism. It binds to a cryptic pocket near the Furin active site, which is not part of the catalytic triad. This binding induces a conformational change in Furin, specifically causing a flip in the W254 residue. This flip creates a new binding pocket that the dichlorophenyl moiety of BOS-318 fills, effectively modulating the enzyme's activity indirectly and selectively. This allosteric mechanism allows BOS-318 to offer a highly selective inhibition of Furin, which could be advantageous in therapeutic contexts where precise modulation of Furin activity is necessary without broadly affecting other proteases [31,48]. Another example of allosteric Furin inhibitors is offered by Permethrin, a recently identified compound that acts through a novel non-competitive allosteric mechanism [44]. Both BOS-318 and Permethrin provide unique perspectives on allosteric inhibition, each with a distinct interaction pattern with Furin, thus serving as useful tools in the development and analysis of new Furin inhibitors. While the well-described BOS-318 could serve as a valuable control in studies of allosteric inhibition of Furin, particularly when compared to P3 inhibitors, Permethrin is also an attractive control. This is because Permethrin may interact with the same Furin pocket targeted by the P3 inhibitor. The similarity in their binding sites can provide important insights into the comparative efficacy and selectivity of these inhibitors. This approach may support future studies aimed at understanding how different molecules can influence Furin function through similar or distinct allosteric mechanisms.
Considering the specificity of Furin substrates influenced by distinct amino acid sequences around the scissile bond, the allosteric inhibition could offer a targeted approach to modulate Furin activity without broadly affecting all its physiological functions. This method might allow for more selective inhibition, potentially reducing the risk of side effects associated with broader enzymatic suppression. Allosteric regulation could thus provide a nuanced control mechanism, offering benefits over traditional active-site inhibitors by potentially maintaining the enzyme physiological roles while selectively inhibiting pathological processing events. This insight is pivotal, aligning

page2image601099728 page2image601099520

with a paradigm shift that prioritizes the inhibition of proteases without fully suppressing their enzymatic activities in vivo, thus avoiding potential detrimental effects.
Reviewer 2: Authors should address more about the effect of mutations to the SARS-CoV2 furin substrate residues as indicated in previous literature in the introduction, this would provide important context and enhance the understanding of the study's findings.

Thank you for your insightful comments regarding the need to elaborate on the effects of mutations to the SARS-CoV-2 furin substrate residues. We acknowledge the importance of this aspect and its relevance to our study findings. As highlighted, our previous publication (Cassari L et al. 2023, https://doi.org/10.3390/ijms24054791) comprehensively explored the influence of these mutations on Furin cleavage efficiency, which underpins the selection of the Omicron SARS peptide sequence as a substrate in the current research.

To address your suggestion directly and enhance the manuscript, we are incorporating a sentence in the discussion section that links back to these foundational findings, thereby providing the necessary context and strengthening the overall understanding of the implications of our current study. We believe this addition will enrich the manuscript by drawing a clear connection between past and present findings.

Revised manuscript, lines 363-380 (blue part):
Of note, during early spread, the pathogen showed a marked propensity to refine those amino acids located all around the S1/S2 boundary, without altering the RARR685 ̄ motif. Specifically, the most popular Omicron variant bears the N679K, P681H replacements that we [12] and others [13] have showed to confer a gain-of-function.

Building on these observation, we designed an extended fluorogenic substrate - SYQTQTKSHRRAR-AMC (SARS peptide) – to be used for in vitro Furin activity assays and inhibitor screenings. This sequence was found to possess excellent cleavability

[12]

Reviewer 2: I would like the authors to comment on the publication: https://doi.org/10.1371/journal.ppat.1009246
Answer: Thank you for your suggestion to consider the study you mentioned. While this research provides insightful findings, our manuscript primarily focuses on newer studies that use pathogen- relevant cell models, offering more directly applicable insights into SARS-CoV-2 dynamics (See our Ref. 20, Essalmani et al. 2022).

The study by Papa et al. (2021) delves into the role of Furin in the processing of the SARS-CoV-2 spike glycoprotein. Using HEK293T cells that stably express ACE2, the authors found that " Processing of the S protein multibasic site is essential for cell-cell fusion but furin is dispensable " They demonstrate that upon depletion of Furin in HEK293T cells, S protein processing is reduced by 80% compared to the wild-type parental cell line. The authors conclude that " This indicates that S protein processing can happen independent of furin, but that the presence of the protease strongly enhances cleavage." These observations align with the findings of a subsequent study by Essalmani R. et al. (DOI: https://doi.org/10.1128/jvi.00128-22), which in addition underscores the cell- dependent importance of Furin. In particular, Essalmani R. et al. state that "Processing of SARS- CoV-2 S by furin-like convertases and TMPRSS2 is critical for viral entry in human lung epithelial

page3image601295088 page3image601295504

The presence of additional positive charges beyond the strictly

conserved RRAR motif significantly enhances the cleavage of the envelope glycoprotein.

Interestingly, although the amino acids at positions 679 and 681 do not directly contact the catalytic

pocket of the enzyme, their specific identities can substantially influence the rate of processing.

This effect is likely mediated through interactions with the surrounding surface area of Furin. The

realization that protagonists of the cleavage extend beyond the 2-4 basic residues near the scissile

bond and the catalytic pocket is crucial. Indeed, understanding that surrounding amino acids can

influence cleavage provides researchers with an additional strategy to interfere with the process,

either by directly interacting with these distant amino acids or by subtly altering the local protease

conformation to induce allosteric effects.

cells but not in model HEK293 cells stably expressing ACE2." Therefore, while the work of Papa et al. (2021) contributes valuable insights, it is essential to complement this research with studies in pathogen-relevant cell lines, such as Calu-3, which more accurately represent the human airway epithelium—a primary site of early SARS-CoV-2 infection. From there, the virus can quickly spread to other tissues and cells like the gut, liver, endothelial cells, and macrophages, where ACE2, furin, and TMPRSS2 are coexpressed (https://doi.org/10.26508/lsa.202000786, https://doi.org/10.1016/j.cell.2020.02.052, https://doi.org/10.1038/s41586-020-2196-x). This comprehensive approach is crucial for a deeper understanding of the dynamics of SARS-CoV-2 infection and the potential therapeutic targets within these processes.

Reviewer 2 Line 226-227 indicates towards Table 4 but the manuscript seems to have only 2 Tables.
Thank you for your attention to detail. You are correct about the discrepancy regarding Table 4 in lines 226-227 of our manuscript. We have revised the manuscript to correct this error and ensure all references to tables are accurate. Thank you for bringing this to our attention, and we appreciate your thorough review.

Old version, lines 226-227
Compounds capable of lowering Furin activity >50% were considered as a hit (Table 4). No potent inhibitors stemmed from the commercially available library (Fig. 4).
Revised manuscript, lines 230-232:
Compounds capable of lowering Furin activity >50% were considered as a hit (Table S2). No potent inhibitors stemmed from the commercially available library (Fig. 4).

Round 2

Reviewer 1 Report

Comments and Suggestions for Authors

The authors have made the required changes, and the manuscript is acceptable for publication in this form.